# Effects of Milk Protein in Resistance Training-Induced Lean Mass Gains for Older Adults Aged ≥ 60 y: A Systematic Review and Meta-Analysis

**DOI:** 10.3390/nu13082815

**Published:** 2021-08-17

**Authors:** Ling-Pi Huang, Giancarlo Condello, Chia-Hua Kuo

**Affiliations:** 1Laboratory of Exercise Biochemistry, Institute of Sports Sciences, University of Taipei, No. 101, Zhongcheng Rd. Sec. 2, Shilin Dist., Taipei 11153, Taiwan; beahuang@yahoo.com (L.-P.H.); giancarlo.condello@gmail.com (G.C.); 2Department of Medicine and Surgery, Anatomy Unit, University of Parma, Via Gramsci 14, 43126 Parma, Italy

**Keywords:** protein supplements, weight training, aging, frailty, sarcopenia, resistance training

## Abstract

This review evaluated the effects of milk-based protein supplementation on resistance training (RT)-induced gains in lean body mass or fat free mass (LBM/FFM) and muscle strength for older adults. A systematic search of PubMed, Scopus and EBSCOhost/SPORTDiscus was conducted. Eligibility criteria: Randomized controlled trials comparing all types of milk-based protein supplements with control supplements for the training older adults at mean age ≥ 60 y. Twenty studies were included in the qualitative synthesis, whilst seventeen studies were included in the quantitative synthesis. A dose of 10–15 g of milk protein supplementation was sufficient to augment RT-induced LBM/FFM. Intriguingly, four out of five studies show negative effect of whey protein supplementation at the same dose range (or even higher) compared with control supplementation (−0.49 kg, 95% CI: −0.69, −0.29, I^2^ = 14%, Z = 4.82, *p* < 0.001). For milk-based protein supplementation, RT-induced improvements in muscle strength were observed only when the protein doses ≥22 g (+0.66 kg, 95% CI: 0.07, 1.25, I^2^ = 0%, Z = 2.18, *p* = 0.03). Conclusion: Milk protein is superior to whey protein in enhancing RT-induced LBM/FFM gains for older adults. Optimal daily protein intake can dilute the protein supplementation effect.

## 1. Introduction

After 60 y of age, muscle mass decreases by an annual rate of 3% [1]. This progressive loss of muscle mass together with weakening muscle strength is significantly associated with mortality [2,3,4,5,6,7,8,9]. Resistance training (RT) has been shown to increase LBM/FFM and muscle strength for older adults including nonagenarians [10,11,12]. Protein is a major macronutrient essential for maintaining contractile components in muscle. However, previous meta-analyses on whether protein supplementation improves RT-induced LBM/FFM gains and muscle strength for older adults present mixed results [13,14,15,16]. This inconsistency in the anabolic responses may be explained by the insufficient amount of protein supplementation [17] and types of the protein with different branched-chain amino acids (BCAA) profile [18]. Furthermore, training status (previously trained with RT) seems to influence the rate of muscle protein turnover [19]. To minimize large heterogeneity among the studies, rigorous control of inclusion criteria is required.

Milk protein consists of a relatively higher amount of BCAA compared with other commonly consumed dietary protein sources [20]. Leucine is the most important amino acid of BCAA in stimulating the muscle protein synthesis [21]. The digestion/absorption property of the protein source can also influence the appearance rate of leucine in circulation after an oral supplementation in older adults with impaired gastrointestinal function [22]. Therefore, milk-based protein products have been considered as the superior protein source for stimulating muscle protein synthesis relative to other dietary protein sources such as soy (8.5% leucine) or beef (8.1% leucine). Among the major protein components of milk protein, whey protein contains relatively higher leucine content (12.3% leucine) than casein [22]. Existing literature comparing the effects of milk protein and whey protein on RT-induced LBM/FFM gains for older adults is rarely available.

A recent meta-analysis has shown that milk protein supplementation alone (without considering training habit) significantly increased appendicular muscle mass in middle-aged adults [23]. However, it remains unclear whether milk protein supplementation promotes RT-induced increases in LBM/FFM and muscle strength for older adults. Milk protein supplementation for RT has not been considered a strict inclusion criterion in the previous meta-analysis [23]. In the present review, studies using trained participants were excluded from the present study to minimize heterogeneity. We aimed to provide evidence-based recommendation for the effect of milk-based protein on RT-induced LBM/FFM gains and muscle strength among older adults aged ≥ 60 y. In addition, subgroup analysis was also conducted to compare the efficacy between milk protein and whey protein on the same outcomes.

## 2. Materials and Methods

This review was conducted according to the Preferred Reporting Items for Systematic Reviews and Meta-Analysis (PRISMA) [24] and was registered on PROSPERO international prospective register for systematic reviews (registration number: CRD42020205299; 20 September 2020). The meta-analysis was performed following the PRISMA guidelines and Cochrane Collaboration handbook [25].

### 2.1. Search Strategy and Study Selection

A systematic literature search was conducted using the online databases and their related thesaurus Medline (PubMed), Scopus, and EBSCOhost/SPORTDiscus for the period from January 2000 to December 2019. The literature search was conducted using the following keywords, as free text terms and thesaurus terms: (aging OR ageing OR older adults OR elderly) AND (protein OR milk OR casein OR whey) AND (resistance OR strength) AND (training OR exercise) AND muscle. In addition, the reference lists of the included studies were reviewed in order to identify other eligible articles.

The literature search was performed independently by two reviewers (LPH and GC) and inconsistencies were solved by consensus. Titles and abstracts generated by the literature search were firstly reviewed. Abstracts without enough information regarding the eligibility criteria were retrieved for full-text evaluation. Full-text articles for those potentially eligible included in the systematic review were obtained and were subsequently screened for relevance using the eligibility criteria.

### 2.2. Eligibility Criteria

Studies were included if they met the following inclusion criteria: (1) full-length, peer-reviewed studies published in English; (2) randomized controlled trials (RCT) on human participants that lasted at least 6 weeks exploring the effect of the combined milk-based protein or non-protein control/placebo supplementation during RT; (3) untrained female and male participants with a mean age of ≥60 y; (4) milk-based protein supplementation, including milk protein, dairy protein, casein, whey, or combinations of whey and essential amino acids, whey and leucine, whey and cysteine, bovine colostrum isolate, concentrate, or hydrolysate consumed in isolation or in combination; (5) RT designed for muscle hypertrophy following the American College of Sports Medicine (ACSM) recommendations for older individuals (i.e., 60–80% of 1 RM for 8–12 repetitions with 1–3 min of rest in between sets for 2–3 day/week, or exercising to volitional fatigue, using both multiple- and single-joint exercises) [26]; (6) any measurement of muscle/lean body mass (by DXA or MRI), and muscle strength.

The following exclusion criteria were considered: (1) the intervention aimed to treat a specific disease or medical condition; (2) co-ingestion of protein supplementation with other potentially hypertrophic agents (e.g., creatine, β-HMB, or testosterone-enhancing compounds); (3) no information regarding the participants’ mean age.

### 2.3. Data Extraction

Using a standardized assessment sheet, two investigators (LPH and GC) independently extracted relevant data: study identifiers (i.e., author identification, year of publication, country of study), participants’ characteristics (i.e., number, age, sex, and body mass), protein supplementation (i.e., type, dose, timing), RT characteristics (i.e., exercise mode, exercise volume and intensity), placebo/control information (i.e., type, dose, timing, or exercise mode, exercise volume and intensity), body composition outcomes (i.e., changes in lean body mass/muscle mass), muscle strength outcomes (i.e., changes in one-repetition-maximum strength).

### 2.4. Assessment of Risk of Bias

The Cochrane Collaboration’s risk of bias tool [25] was used to assess the risk of bias of the included studies. This tool evaluates the random sequence generation, allocation concealment, blinding of participants and personnel, blinding of outcome assessment, incomplete outcome data, selective outcome reporting, and others which are not covered in the above. Each study was labelled as either a low risk of bias, a high risk of bias, or an unclear risk of bias. The data included in the meta-analyses were restricted to studies with less than two reported high-risk domains.

### 2.5. Statistical Analysis

The random-effects meta-analysis was performed using the Review Manager software (RevMan 5.3; Cochrane Collaboration, Oxford, UK). Mean differences (MD) and 95% confidence intervals (95% CIs) were calculated for muscle strength and body composition outcomes. Only pre-intervention and post-intervention outcome data were retrieved for treatment and control groups. If a study had multiple measures of outcomes (e.g., handgrip strength and leg extension strength), data from a lower extremity were retrieved. If data were not presented in the study, data were calculated from baseline values and/or percentage change. If Δ*SD* was not reported, the correlation coefficient for each primary outcome was calculated according to the Cochrane Handbook for Systematic Reviews of Interventions [27]:$$corr=\left(SDpre^{2}+SDpost^{2}-SDchange^{2}\right)/\left(2\times SDpre\times SDpost\right)$$
and the *SD* was then calculated as:$$∆SD=\sqrt{\left(SDpre^{2}+SDpost^{2}-2\times corr\times SDpre\times SDpost\right)}$$

Subgroup analyses were performed using the subgroup analysis function within the Review Manager software (RevMan 5.3; Cochrane Collaboration, Oxford, UK) on: (1) daily dietary protein intake; (2) protein supplementation dosage; and (3) protein supplementation type.

Heterogeneity among studies was evaluated through I^2^ statistics, the Cochrane Chi square (χ^2^), and the between-study variance using the tau-square (τ^2^). The heterogeneity thresholds were I^2^ = 0% to 40% (might not be important), I^2^ = 30% to 60% (may represent moderate heterogeneity), and I^2^ = 50% to 90% (may represent substantial heterogeneity), I^2^ = 75% to 100% (considerable heterogeneity) [25]. A *p* value < 0.1 for χ^2^ was defined as indicating the presence of heterogeneity. A τ^2^ > 1 suggested the presence of substantial statistical heterogeneity. The level of statistical significance was set at *p* < 0.05, whilst a moderate significance was declared as *p* = 0.05–0.10.

## 3. Results

### 3.1. Literature Search

The initial electronic search identified 1574 articles potentially eligible for inclusion, together with 10 articles identified through the reference lists, resulting in a total of 1584 articles. After the removal of 376 duplicated articles, 1208 articles were screened based on title and abstract. In total, 56 full-text articles were assessed, but 36 of them were then excluded due to not complying the eligibility criteria. As the result, a total of 20 studies met the inclusion criteria and were included in the systematic review for qualitative analysis, whilst 17 studies were included in the meta-analysis for quantitative analysis (Figure 1). Three studies [28,29,30] were not included in the meta-analysis due to insufficient data.

All 20 included studies were randomized controlled trials with a placebo group. These studies were conducted in 10 different countries with the majority of intervention programs lasting 12 weeks (ranged from 10 weeks to 18 months). Participants from 14 studies were either single or double blinded [28,29,31,32,33,34,35,36,37,38,39,40,41,42], 6 were non-blinded [30,43,44,45,46,47]. Detailed studies’ characteristics are reported in Table 1.

### 3.2. Risk of Bias Assessment

Risk of bias assessment for each included study is presented in Figure 2. High risk of bias was evident only for selective reporting bias in 6 studies [35,40,41,44,45,46], as their methods of measuring the outcomes may not be appropriate. Muscle mass measured by bioelectrical impedance analyzer in two studies [46,47] were rated high risk and were excluded from the quantitative analysis. Skeletal muscle derived from lean mass was also rated high risk but were included in the quantitative analysis as the equation is constant [40,41].

### 3.3. Participants Characteristics

The total number of participants across all studies was 1544, with age ranging from 50 to 91 years and BMI ranging from 22 to 31 kg/m^2^. Studies either used a mixed-sex sample [30,31,32,33,34,37,38,42,44,46,47] or a men-only sample [28,29,35,36,39,43] and women-only sample [40,41,45]. Participants were declared to be “healthy” in nine studies [29,30,34,35,39,40,43,44,45], having a condition of sarcopenia, frailty or limited mobility in seven studies [31,33,36,37,38,42,47], and healthy and with hypertension, hyperlipidemia, or type 2 diabetes in one study [32].

### 3.4. RT Characteristics

The RT interventions lasted from 10 weeks to 18 months (29.7 ± 25.7 weeks) performing RT from 2 to 5 days per week (2.9 ± 0.8 days/week), with 2 to 14 exercises per session (8.0 ± 2.9 exercises/session), with 2 to 4 sets per exercise (2.7 ± 0.8 sets/exercise), with 8 to 20 repetitions per set (13.9 ± 4.2 repetitions/set) (or to fatigue), and intensity at 50–85% 1RM. Two studies performed only lower-body RT [29,38], whilst the rest involved whole-body exercises.

### 3.5. Effect of Daily Protein Intake on the RT-induced LBM/FFM Gains

A range of 10 to 40 g of protein doses was orally received in the protein supplemented group including milk, whey, and casein protein along with RT. Eight studies supplied protein supplements after RT section [28,29,32,36,40,41,42,45]. In total, 12 studies provided protein supplements daily in addition to their regular meals [30,31,33,34,35,37,38,39,43,44,46,47], whilst 5 of these studies provided the protein supplements after RT and other time in addition to regular meals [33,35,38,39,44]. Among those protein supplemented group, their reported daily protein intake ranged from 0.85 to 1.39 g/kg/day prior intervention, whilst ranged from 0.90 to 1.53 g/kg/day during intervention resulting in a significant increase in average daily protein intake of 0.35 g/kg/day (*p* < 0.001). All control groups did not receive protein supplementation with RT. For those control group, daily protein intake prior intervention ranged from 0.81 to 1.32 g/kg/day, whilst ranged from 0.80 to 1.31 g/kg/day during intervention, without a significant change by the interventions (−0.07 g/kg/day, *p* = 0.72).

### 3.6. Effect of Milk-Based Protein Supplementation on LBM/FFM

Milk-based protein supplementation may enhance RT-induced gains in LBM/FFM according to 14 studies encompassing 972 participants with high heterogeneity (+0.31 kg, 95% CI: 0.00, 0.62, I^2^ = 88%, overall effect: Z = 1.96, *p* = 0.05) (Figure 3). To address this heterogeneity of the included studies, subgroup analyses was conducted. The result reveals a greater effect on RT-induced gains in LBM/FFM when a daily dietary protein intake (excluding protein supplements for training) ≤1.1 g/kg/day (+0.45 kg, 95% CI: 0.17, 0.72, I^2^ = 65%, overall effect: Z = 3.22, *p* = 0.001) (Figure 4). Subgroup analysis for the dose of protein supplements along with RT also showed a significant effect when the protein ≥22 g (+0.62 kg, 95% CI: 0.26, 0.97, I^2^ = 67%, overall effect: Z = 3.38, *p* = 0.0007) (Figure 5). Furthermore, subgroup analysis for supplemented protein type demonstrated a significant effect of milk-based protein supplementation only for the milk protein (+0.43 kg, 95% CI: 0.18, 0.68, I^2^ = 41%, overall effect: Z = 3.34, *p* = 0.0008) but not for the whey protein (Figure 6). The effect of whey protein was significant only when a protein dose ≥22 g (+0.57 kg, 95% CI: 0.19, 0.95, I^2^ = 66%, overall effect: Z = 2.95, *p* = 0.003) (Figure 7).

### 3.7. Effect of Milk-Based Protein Supplementation on Muscle Strength

Milk-based protein supplementation may enhance RT-induced gains in 1RM for knee extension according to 10 studies with 638 participants (+0.32 kg, 95% CI: −0.04, 0.68, I^2^ = 0%, overall effect: Z = 1.72, *p* = 0.09). Subgroup analysis for protein supplementation dosage demonstrated a significant effect only for a protein dose ≥22 g (+0.66 kg, 95% CI: 0.07, 1.25, I^2^ = 0%, overall effect: Z = 2.18, *p* = 0.03) but not for a dose <22 g (Figure 8).

## 4. Discussion

This review provides a new summary of evidence regarding the effects of milk-based protein supplementation on RT-induced gains in LBM/FFM and muscle strength for older adults with mean age ≥ 60 y, the population prone to muscle loss [48]. The main findings of the study are (1) Milk protein supplements are more effective than whey protein supplements on RT-induced increases in LBM/FFM for the older adults aged ≥ 60 y; (2) For the overall effect, milk-based protein (all types of protein produced from milk) supplementation moderately augments RT-induced increases in LBM/FFM and muscle strength in the older adults aged ≥ 60 y. Evidence from previous meta-analyses regarding the enhancement effect of protein supplementation on RT-induced gains in LBM/FFM and muscle strength for the age level was contradictory [13,14,15,16]. The discrepancies are likely caused by the differences in inclusion criteria of each meta-analysis, such as the type and dose of protein supplementation, the type of exercise or training volume, and the training status of participants. In this meta-analysis, rigorous inclusion criteria were set to restrict the search strategy to avoid the increase in the variability of findings. The supplementation has been limited to milk-based protein, the participants were previously untrained, and the RT has to be designed to induce skeletal muscle hypertrophy.

### 4.1. Effect of Milk-Based Protein Supplementation Type

Gorissen and colleagues [49] found a relatively higher amino acid appearance in circulation 5 h following consumption of milk protein compared with whey protein and casein protein, reflecting a superior amino acid bioavailability of milk protein than whey protein. In the present study, evidence from subgroup analysis for protein supplementation type indicates that milk protein supplementation is more effective than whey protein on RT-induced increases in LBM/FFM for older adults with mean age ≥ 60 y (Figure 6). Whey protein supplementation was less effective than the control supplementation on RT-induced gains in LBM/FFM when the dosage were <22 g (Z = 4.82, *p* < 0.001) (Figure 7). Among the 8 studies in the subgroup, whey protein supplementation to achieve significant RT-induced gains in LBM/FFM requires protein dosage ≥22 g (Z = 2.95, *p* = 0.003) (Figure 9). Since timing of protein supplementation is crucial for resistance training-induced LBM/FFM gains for young men [50], whey protein is generally considered as the optimal protein source to support RT-induced LBM/FFM gains due to its fast release into circulation after digestion [51]. Casein and some other nutrient components are largely excluded in the whey protein supplements. Therefore, the result of the present meta-analysis implicates the importance of nutrient diversity and sustaining amino acid release after digestion on maintaining RT-induced LBM/FFM for older adults [52]. Furthermore, 7 out of the 8 whey protein studies used an isocaloric carbohydrate as the control group. It is generally known that carbohydrate has stronger action in stimulating pancreatic insulin release into circulation than protein. Insulin is a potent anabolic hormone which inhibits muscle protein breakdown and DNA synthesis (for cell regeneration) [53,54]. Since insulin secretion decreases during old age in humans [55], this age dependent factor might explain the negative outcome in some of the whey protein studies using carbohydrate as control supplements for the older adults with mean age ≥ 60 y.

The results of the subgroup analysis suggest that milk protein is a superior protein source to whey protein in RT-induced LBM/FFM gains for older adults with mean age ≥ 60 y. For the milk protein subgroup, the protein dosage of the six studies included in the subgroup analysis ranges from 10 to 30 g. The overall effect of the milk protein subgroup was significant. Among the six studies, four studies [34,36,43,46] provided protein doses ≤15 g (ranged 10–15 g) (Figure 9) demonstrating a positive effect for RT-induced LBM/FFM gains (MD = 0.31 kg, 95% CI: 0.15, 0.47, I^2^ = 0%, Z = 3.76, *p* = 0.0002), whereas whey protein requires >22 g to obtain a significant effect. The amount of 10–15 g of milk protein is roughly equivalent to the proteins content from two servings of milk (1 serving of milk = 240 mL, protein 8 g) [56].

### 4.2. Effect of Milk-Based Protein Supplementation on RT-Induced LBM/FFM Gains

The moderate effect (Z = 1.96, *p* = 0.05) of milk-based protein supplementation (all types of protein produced from milk) on RT-induced gains in LBM/FFM is associated with a large heterogeneity among studies. Subgroup analysis indicates that daily dietary protein intake (excluding protein supplements given with training) is a major confounder for the effect of milk-based protein supplementation on RT-induced gains in LBM/FFM. Similar to the present study, a dilution effect of daily protein intake on protein supplementation for RT-induced LBM/FFM gains has been reported elsewhere using different inclusion criteria [17]. In the present meta-analysis, 9 studies included in the subgroup of daily protein intake limiting to ≤1.1 g/kg/day, as a cut-off point based on previous study [57], reported a greater overall effect in RT-induced LBM/FFM gains (Z = 3.22, *p* < 0.001). This positive effect is contributed by 4 studies showing a significant difference between the supplemented and control groups [31,37,40,41] and 4 studies showing a similar trend without obtaining a statistical significance. One study from Arnarson and colleagues showed no difference (towards negative effect) between milk-based protein and control groups [32]. A possible explanation for this exception is probably associated with compensating protein intake, indicated by unchanged total protein intake (supplement + meal) for the supplementation group (1.00 to 1.06 g/kg/day) before and after the intervention. Their observation is consistent with a study included in qualitative analysis (not included in the meta-analysis due to absence of absolute values in LBM/FFM) [29].

Among the 5 studies included in the subgroup limiting to daily protein intake >1.1 g/kg/day, 4 studies [34,39,42,43] showed no effects of milk-based protein supplementation on RT-induced gains in LBM/FFM. Only 1 study showed significant LBM/FFM gains [46]. In that exception, the protein dose was low (10 g) and other micronutrients such as zinc, vitamin B12, folic acid, and vitamin D were also included in the supplemented group but not in the control group [46]. More studies would be needed to clarify the potential effect of other micronutrients on protein supplementation in RT-induced LBM/FFM gains.

Recommendations of daily dietary protein intake for older adults are varied according to different studies. The cut-off value of 1.1 g/kg/day of daily protein intake for older adults provided by the present meta-analysis is in line with the contribution provided by the PROT-AGE study group which recommends an average daily intake range of 1.0–1.2 g/kg/day to maintain and regain muscle mass regardless of training habit [57]. Result of the analysis also fits well with European recommendation of protein intake for elderly ranges from 1.0 to 1.5 g per day [58].

The dosage of protein supplementation has been found to influence the effect of milk-based protein supplementation on RT-induced gains in LBM/FFM. A clear positive effect was observed when a protein dose ≥22 g of all milk-based protein supplements for older adults conducting RT, regardless of protein types. Scatter plot for protein dosage and LBM/FFM gains (Figure 9) indicated a trend in favor of RT-induced LBM/FFM gains in 6 studies included in the ≥22 g subgroup and 4 of the 6 studies show significant effects [31,37,40,41]. All these 6 studies were also included in the subgroup with daily dietary protein intake ≤1.1 g/kg/day excluding protein supplemented with training. These studies support the notion that supplementing a larger amount of milk-based protein to older adults with inadequate dietary protein intake has a larger effect in increased RT-induced gains in LBM. Data from of Chalé et al. [33] and Thomson et al. [44] show insignificant increases in LBM/FFM (1.3% and 1%, respectively) for the protein supplemented group and in the control group (0.6% and 0.8%, respectively). Authors explained that the insignificant result may be associated with a relatively lower adherence (72%) [33]. Thomson and colleagues explain the insignificant effect in increased RT-induced LBM/FFM gains in LBM to be associated with low training volume [44].

It has been reported a lower appearance of dietary protein-derived phenylalanine in the circulation at 5-h postprandial period in older adults compared with young adults [49]. Greater protein dose is required for older adults to maximize muscle protein synthesis compared with younger adults [48]. Therefore, lack of effects on RT-induced LBM/FFM gains for the older adults with the protein dose <22 g [38,39,42] is probably associated with poor digestive function and loss of anabolic function in older adults.

### 4.3. Effect of Milk-Based Protein Supplementation on RT-Induced Muscle Strength

The overall effect of milk-based protein supplementation on RT-induced gains in 1RM for knee extension was found to be moderate, therefore conclusive evidence cannot be obtained from the overall analysis. However, the subgroup analysis suggests a significant benefit of milk-based protein supplementation when protein supplementation dosage was ≥22 g. Similarly, the studies in the qualitative analysis also showed significant RT-induced increases in knee extension strength [45] and leg press, chest press, and bicep curl strength [30] at similar protein doses. The participants from those studied have reported an average daily protein intake > 1.1 g/kg/day. However, previous meta-analyses provided contrasting results, with 2 meta-analyses showing a positive effect of protein supplementation combined with RT on muscle strength [13,16] and another 2 systematic review/meta-analyses failed to confirm the positive effects on RT-induced increases in muscle strength [14,15]. The contradictory effects might be attributed to different inclusion criteria and outcome measured.

### 4.4. Limitations

This review has some limitations which need to be considered to cautiously interpret the results. First, we could not preclude the possibility that other milk-based non-protein components are contributor to the RT-induced LBM/FFM gains for the older adults, particularly in the explanation of superiority of milk protein to whey protein [59]. Second, the eligibility criteria defined in the meta-analysis restricts the number of included studies. For example, only untrained individuals were included in the analysis. It has been previously suggested that resistance-trained individuals might need higher dietary intake and protein supplementation [17,54].

## 5. Conclusions

Milk protein supplementation is more effective than whey protein supplementation in RT-induced LBM/FFM gains for the older adults with mean age ≥ 60 y (Milk protein: +0.43 kg, 95% CI: 0.18, 0.68, Z = 3.34, *p* < 0.001; Whey protein: +0.15 kg, 95% CI: −0.33, 0.62, Z = 0.60, *p* = 0.55). The positive effect of milk protein supplementation on RT-induced LBM/FFM gains can be observed when the amount of protein is within the range of 10–15 g of milk protein, whereas whey protein supplementation at the same or even higher doses is less effective than control supplementation. This may explain the overall moderate effect of milk-based protein supplementation (all protein products from milk including whey protein) on augmenting the RT-induced LBM/FFM gains (+0.31 kg, 95% CI: 0.00, 0.62, Z = 1.96, *p* = 0.05) and muscle strength (+0.32 kg, 95% CI: −0.04, 0.68, Z = 1.72, *p* = 0.09). A daily dietary protein intake (excluding supplementation during training) beyond ~1.1 g/kg/day dilutes milk-based protein supplementation effect in RT-induced increases in lean mass in older adults. Given the difference in amino acid sustainability in circulation between milk protein and whey protein, the result of this meta-analysis encourages the protein supplementation regimen aiming to maintain circulating amino acid availability for training individuals aged ≥ 60 y. Two cups of milk a day which provides 16 g of milk protein, combined with resistance training, could be considered an effective strategy against loss of muscle mass and strength at this age level.

## Figures and Tables

**Figure 1 nutrients-13-02815-f001:**
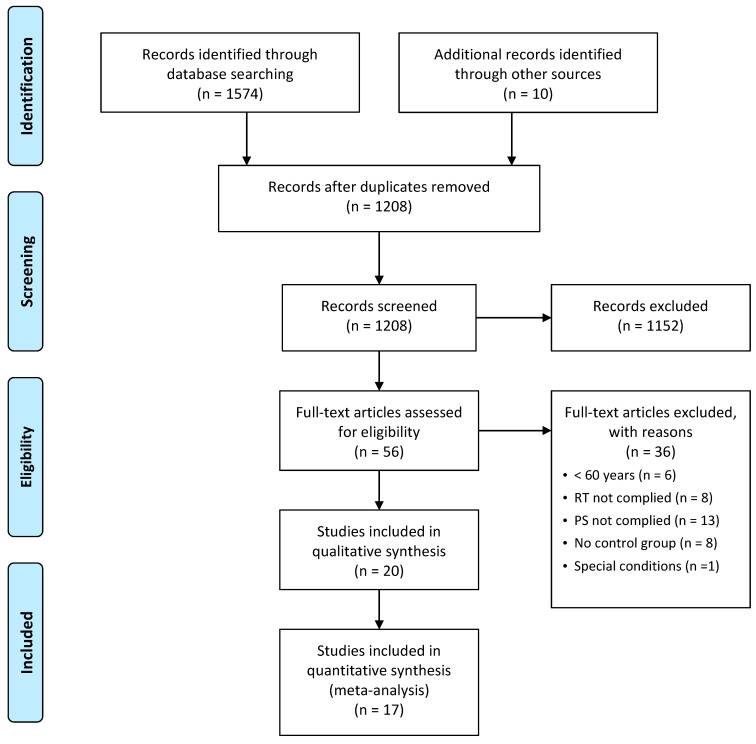
Flow diagram of the literature search and selection process. PS protein supplementation; RET resistance exercise training.

**Figure 2 nutrients-13-02815-f002:**
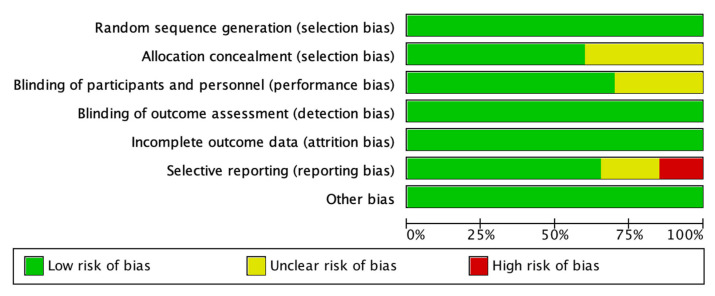
Risk of bias assessment presented as a percentage across all included studies.

**Figure 3 nutrients-13-02815-f003:**
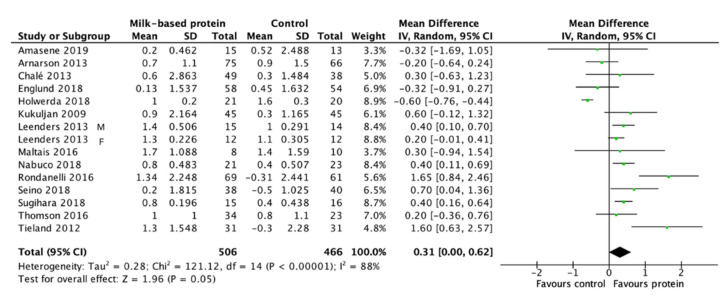
Forest plot for the differences in RT-induced LBM/FFM gain between consuming milk-based protein supplements and control supplements above age 60 years.

**Figure 4 nutrients-13-02815-f004:**
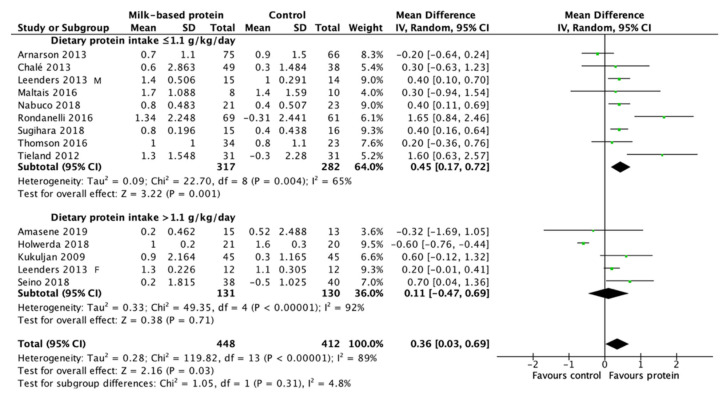
Forest plot for the differences in RT-induced LBM/FFM gain between consuming milk-based protein supplements and control supplements considering the subgroup analysis of daily dietary protein intake.

**Figure 5 nutrients-13-02815-f005:**
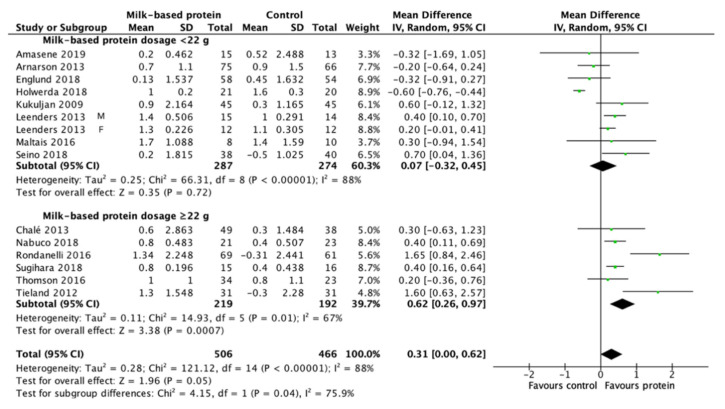
Forest plot for the differences in RT-induced LBM/FFM gain between consuming milk-based protein supplements and control supplements considering the subgroup analysis of protein supplementation dosage.

**Figure 6 nutrients-13-02815-f006:**
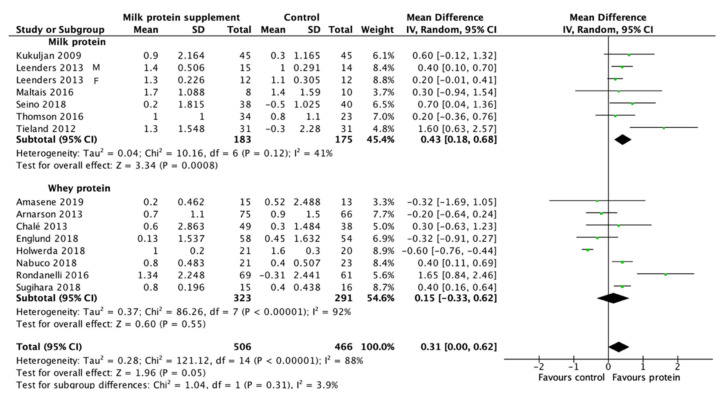
Forest plot for the differences in RT-induced LBM/FFM gain between consuming milk-based protein supplements and control supplements considering the subgroup analysis of protein supplementation type.

**Figure 7 nutrients-13-02815-f007:**
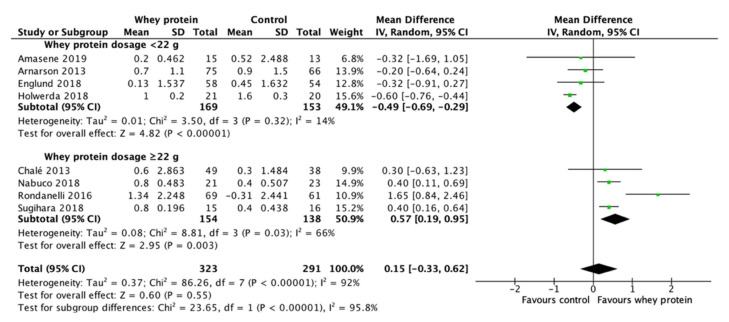
Forest plot for the differences in RT-induced LBM/FFM gain between consuming whey protein supplements and control supplements considering the subgroup analysis of protein dosage.

**Figure 8 nutrients-13-02815-f008:**
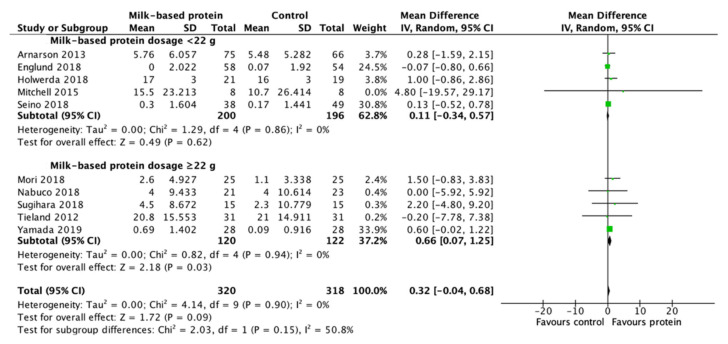
Forest plot for the differences in RT-induced increases in 1RM knee extension between consuming milk-based protein supplements and control supplements considering the subgroup analysis of protein dosage.

**Figure 9 nutrients-13-02815-f009:**
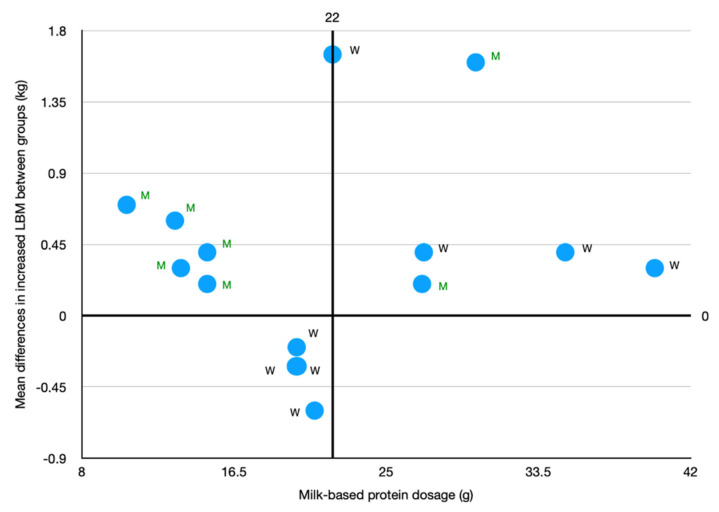
Relationship between milk-based protein dosage and the difference between milk-base protein and control supplementations on RT-induced LBM/FFM gain. Each blue dot represents a single study. M: milk protein; W: whey protein.

**Table 1 nutrients-13-02815-t001:** Characteristics of the intervention studies.

**Study, Year** **Country**	**Intervention** **Duration**	**Number of Participants**	**Participant** **Characteristics**	**Resistance** **Training**	**Protein** **Supplementation**	**Protein Intake** **Pre (Post) Intervention**	**Outcome** **Measured**
Amasene et al., 2019 [39] Spain	12 weeks	28	Female and male Post-hospitalization Age: 82 years. BMI: 27–30 kg/m^2^	2 days/week 2 sets to max reps 6 exercises 50–70% 1RM	20 g whey (3 g leucine) post exercise	N/A	LBM
Arnarson et al., 2013 [32] Iceland	12 weeks	161	Female and male Healthy, or hypertension, hyperlipidemia, diabetes type II Age: 65–91 years. BMI: 28–29 kg/m^2^	3 days/week 3 sets × 6–8 reps 10 exercises 75–80% 1RM	20 g whey post RET	P: 1.00 ± 0.26 kg/d (1.06 ± 0.23 kg/d) CON: 0.92 ± 0.30 kg/d (0.89 ± 0.23 kg/d)	LBM KE
Candow et., 2006 [28] Canada	12 weeks	38	Male Age: 59–76 years. BMI: 28.2 kg/m^2^	3 days/week 3 sets × 10 reps 8 exercises 70% 1RM	0.3 g/kg (~25.6 g) whey post RET	P: 1.14 kg/d (1.38 kg/d) CON: 1.28 kg/d	LBM LP
Chale et al., 2013 [33] USA	6 months	80	Female and male Mobility-limited Age: 70–85 years. BMI: 27 kg/m^2^	3 days/week 3 sets × 12 reps 5 exercises 80% 1RM	40 g whey daily post RET	P: 0.96 kg/d (1.22 kg/d) CON: 0.92 kg/d	LBM LP
Englund et al., 2017 [38] USA	6 months	149	Female and male Mobility-limited Age: 78 ± 5.4 years. BMI: 28 kg/m^2^	3 days/week 2 sets × 10 reps 5 leg-exercises 15 to 6 Borg’s scale	20 g whey daily or post RET	N/A	LBM KE
Holwerda et al., 2018 [39] Netherlands	12 weeks	41	Female and male Healthy Age: 70 ± 1 years. BMI: 25.3 ± 0.4 kg/m^2^	3 days/week 2–4 sets × 10 reps 6 exercises 70–80% 1RM	21 g whey protein (3 g leucine) post RET and before sleep	P: 1.14 ± 0.05 kg/d (1.43 ± 0.04 kg/d) CON: 1.19 ± 0.06 kg/d (1.17 ± 0.06 kg/d)	LBM KE
Kirk et al., 2019 [30] United Kindom	16 weeks	46	Female and male Non-frail healthy Age: 68 ± 5 years. BMI: 27.8 ± 6.2 kg/m^2^	2 days/week 2 sets to fatigue 6 exercises moderate weight	1.5 g/kg/day whey 0.5 g × 3 times/day	P: 1.16 ± 0.4 kg/d (1.63 ± 0.5 kg/d) CON: 1.10 ± 0.4 kg/d (1.04 ± 0.3 kg/d)	LP
Kukuljan et al., 2009 [43] Australia	18 months	180	Male Healthy Age: 50–79 years. BMI: 27.1 kg/m^2^	3 days/week 3–4 sets × 15–20 reps 10–14 exercises 60–85% 1RM	13.2 g milk protein daily	P: 1.26 ± 0.32 kg/d (+0.06 kg/d) CON: 1.32 ± 0.32 kg/d (−0.1 kg/d)	LBM
Leenders et al., 2013 [34]	24 weeks	60	Female and male Healthy Age: 78 ± 1 years. BMI: 24–27 kg/m^2^	3 days/week 3–4 sets × 8 reps 4 exercises 50–80% 1RM	15 g milk protein daily after breakfast	Women: 1.2 kg/d P: (+0.24 kg/d) Men: 1.1 kg/d P: (+0.18 kg/d)	LBM LP
Maltais et al., 2016 [36] Canada	16 weeks	26	Male Sarcopenic Age: 65 ± 5 years. BMI: 26–27 kg/m^2^	3 days/week 3 sets × 8 reps8 exercisers 80% 1RM	13.53 g milk protein post RET	P: 1.03 kg/d (2.12 kg/d) CON: 1.25 kg/d (1.06 kg/d)	LBM
Mitchell et al., 2015 [35] Canada	12 weeks	16	Male Healthy Age: 74.4 ± 5.4 years. BMI: 26.9 ± 3.2 kg/m^2^	2 days/week 4 × leg exercises 1 days/week 7 × upper body exercises 3–4 sets 75–80% 1RM	14 g milk protein daily post RET or breakfast	N/A	KE
Mori and Tokuda, 2018 [45] Japan	24 weeks	81	Female Healthy Age: 70.6 ± 4 years. BMI: 22–23 kg/m^2^	2 days/week > 40 min 7 exercises 50–70% 1RM	22.3 g whey post RET	Both groups 1.3 ± 0.0 kg/d (1.4 ± 0.1 kg/d)	LLMM KE
Nabuco et al., 2018 [40] Brazil	12 weeks	70	Female Healthy Age: > 60 years. BMI: 23–26.5 kg/m^2^	3 days/week 3 sets × 8–12 RM 8 exercises	27.1 g whey post RET	P: 0.94 ± 0.36 kg/d (1.49 ± 0.46 kg/d) CON: 0.95 ± 0.27 kg/d (1.0 ± 0.25 kg/d)	SMM KE
Rondanelli et al., 2016 [37] Italy	12 weeks	130	Female and male Sarcopenic Age: 80.3 years. BMI: 23.9 kg/m^2^	5 days/week 20-min/day 12–14 Borg Rate	22 g whey daily	P: 0.9 g/kg (unchanged) CON: 1.0 g/kg (unchanged)	FFM HG
Seino et al., 2018 [46] Japan	12 weeks	82	Female and male Non-disabled Age: 73.5 years. BMI: 22.9 ± 2 kg/m^2^	2 days/week 2 sets × 20 reps whole body somewhat hard	10.5 g milk protein daily	P: 1.39 ± 0.36 kg/d (1.53 ± 0.33 kg/d) CON: 1.28 ± 0.26 kg/d (1.31 ± 0.26 kg/d)	LBM KE
Sugihara Junior et al., 2018 [41] Brazil	12 weeks	31	Female Age: 67.4 ± 4.0 years. BMI: 25.5 ± 2.4 kg/m^2^	3 days/week 3 sets × 8–12 RM 8 exercises	35 g whey post RET	P: 0.85 ± 0.1 kg/d (1.4 ± 0.1 kg/d) CON: 0.81 ± 0.1 kg/d (0.87 ± 0.1 kg/d)	SMM KE
Thomson et al., 2016 [44] Australia	12 weeks	125	Female and male Healthy Age: 61.5 ± 7.4 years. BMI: 27.6 ± 3.6 kg/m^2^	3 days/week 3 sets × 8–12 reps 5 exercises 8RM	27 g dairy protein spread to each main meal or post RET	P: 1.06 ± 0.10 kg/d (1.42 ± 0.14 kg/d) CON: 1.10 ± 0.10 kg/d	LBM KE
Tieland et al., 2012 [31] Netherlands	24 weeks	62	Female and male Prefrailty and frailty Age: 78 ± 1 years.	2 days/week 3–4 sets × 15–20 reps 6 exercises 50–75% 1RM	30 g milk protein (15 g at breakfast and lunch)	P: 1.0 kg/d (1.3 kg/d) CON: 1.0 kg/d	LBM KE
Verdijk et al., 2009 [29] Netherlands	12 weeks	26	Male Healthy Age: 72 ± 2 years. BMI: 26~27 kg/m^2^	3 days/week 4 sets × 8 reps Leg extension and leg press 75–80% 1RM	20 g casein before and post RET	Both groups (unchanged) 1.1 ± 0.1 kg/d (unchanged)	LBM KE LP
Yamada et al., 2019 [47] Japan	12 weeks	112	Female and male sarcopenic or dynapenic Age: 84.2 ± 5.5 years. BMI: 22 kg/m^2^	2 days/week 7 exercises 3 sets × 20 reps	10 g whey protein, daily	N/A	AMM KE
Study, Year Country	Duration	Sample Size	Participant Characteristics	Resistance Training	Protein Supplementation	Daily Protein Intake Pre (Post)	Outcome Measured
Amasene et al., 2019 [39] Spain	12 weeks	28	Female and male Post-hospitalization Age: 82 years. BMI: 27–30 kg/m^2^	2 days/week 2 sets to max reps 6 exercises 50–70% 1RM	20 g whey (3 g leucine) post exercise	N/A	LBM
Arnarson et al., 2013 [32] Iceland	12 weeks	161	Female and male Healthy, or hypertension, hyperlipidemia, diabetes type II Age: 65–91 years. BMI: 28–29 kg/m^2^	3 days/week 3 sets × 6–8 reps 10 exercises 75–80% 1RM	20 g whey post RET	P: 1.00 ± 0.26 kg/d (1.06 ± 0.23 kg/d) CON: 0.92 ± 0.30 kg/d (0.89 ± 0.23 kg/d)	LBM KE
Candow et al., 2006 [28] Canada	12 weeks	38	Male Age: 59–76 years. BMI: 28.2 kg/m^2^	3 days/week 3 sets × 10 reps 8 exercises 70% 1RM	0.3 g/kg (~25.6 g) whey post RET	P: 1.14 kg/d (1.38 kg/d) CON: 1.28 kg/d	LBM LP
Chale et al., 2013 [33] USA	6 months	80	Female and male Mobility-limited Age: 70–85 years. BMI: 27 kg/m^2^	3 days/week 3 sets × 12 reps 5 exercises 80% 1RM	40 g whey daily post RET	P: 0.96 kg/d (1.22 kg/d) CON: 0.92 kg/d	LBM LP
Englund et al., 2017 [38] USA	6 months	149	Female and male Mobility-limited Age: 78 ± 5.4 years. BMI: 28 kg/m^2^	3 days/week 2 sets × 10 reps 5 leg-exercises 15 to 6 Borg’s scale	20 g whey daily or post RET	N/A	LBM KE
Holwerda et al., 2018 [39] Netherlands	12 weeks	41	Female and male Healthy Age: 70 ± 1 years. BMI: 25.3 ± 0.4 kg/m^2^	3 days/week 2–4 sets × 10 reps 6 exercises 70–80% 1RM	21 g whey protein (3 g leucine) post RET and before sleep	P: 1.14 ± 0.05 kg/d (1.43 ± 0.04 kg/d) CON: 1.19 ± 0.06 kg/d (1.17 ± 0.06 kg/d)	LBM KE
Kirk et al., 2019 [30] UK	16 weeks	46	Female and male Non-frail healthy Age: 68 ± 5 years. BMI: 27.8 ± 6.2 kg/m^2^	2 days/week 2 sets to fatigue 6 exercises moderate weight	1.5 g/kg/day whey 0.5 g × 3 times/day	P: 1.16 ± 0.4 kg/d (1.63 ± 0.5 kg/d) CON: 1.10 ± 0.4 kg/d (1.04 ± 0.3 kg/d)	LP
Kukuljan et al., 2009 [43] Australia	18 months	180	Male Healthy Age: 50–79 years. BMI: 27.1 kg/m^2^	3 days/week 3–4 sets × 15–20 reps 10–14 exercises 60–85% 1RM	13.2 g milk protein daily	P: 1.26 ± 0.32 kg/d (+0.06 kg/d) CON: 1.32 ± 0.32 kg/d (−0.1 kg/d)	LBM
Leenders et al., 2013 [34]	24 weeks	60	Female and male Healthy Age: 78 ± 1 years. BMI: 24–27 kg/m^2^	3 days/week 3–4 sets × 8 reps 4 exercises 50–80% 1RM	15 g milk protein daily after breakfast	Women: 1.2 kg/d P: (+0.24 kg/d) Men: 1.1 kg/d P: (+0.18 kg/d)	LBM LP
Maltais et al., 2016 [36] Canada	16 weeks	26	Male Sarcopenic Age: 65 ± 5 years. BMI: 26–27 kg/m^2^	3 days/week 3 sets × 8 reps8 exercisers 80% 1RM	13.53 g milk protein post RET	P: 1.03 kg/d (2.12 kg/d) CON: 1.25 kg/d (1.06 kg/d)	LBM
Mitchell et al., 2015 [35] Canada	12 weeks	16	Male Healthy Age: 74.4 ± 5.4 years. BMI: 26.9 ± 3.2 kg/m^2^	2 days/week 4 × leg exercises 1 days/week 7 × upper body exercises 3–4 sets 75–80% 1RM	14 g milk protein daily post RET or breakfast	N/A	KE
Mori and Tokuda, 2018 [45] Japan	24 weeks	81	Female Healthy Age: 70.6 ± 4 years. BMI: 22–23 kg/m^2^	2 days/week > 40 min 7 exercises 50–70% 1RM	22.3 g whey post RET	Both groups 1.3 ± 0.0 kg/d (1.4 ± 0.1 kg/d)	LLMM KE
Nabuco et al., 2018 [40] Brazil	12 weeks	70	Female Healthy Age: > 60 years. BMI: 23–26.5 kg/m^2^	3 days/week 3 sets × 8–12RM 8 exercises	27.1 g whey post RET	P: 0.94 ± 0.36 kg/d (1.49 ± 0.46 kg/d) CON: 0.95 ± 0.27 kg/d (1.0 ± 0.25 kg/d)	SMM KE
Rondanelli et al., 2016 [37] Italy	12 weeks	130	Female and male Sarcopenic Age: 80.3 years. BMI: 23.9 kg/m^2^	5 days/week 20-min/day 12–14 Borg Rate	22 g whey daily	P: 0.9 g/kg (unchanged) CON: 1.0 g/kg (unchanged)	FFM HG
Seino et al., 2018 [46] Japan	12 weeks	82	Female and male Non-disabled Age: 73.5 years. BMI: 22.9 ± 2 kg/m^2^	2 days/week 2 sets × 20 reps whole body somewhat hard	10.5 g milk protein daily	P: 1.39 ± 0.36 kg/d (1.53 ± 0.33 kg/d) CON: 1.28 ± 0.26 kg/d (1.31 ± 0.26 kg/d)	LBM KE
Sugihara Junior et al., 2018 [41] Brazil	12 weeks	31	Female Age: 67.4 ± 4.0 years. BMI: 25.5 ± 2.4 kg/m^2^	3 days/week 3 sets × 8–12RM 8 exercises	35 g whey post RET	P: 0.85 ± 0.1 kg/d (1.4 ± 0.1 kg/d) CON: 0.81 ± 0.1 kg/d (0.87 ± 0.1 kg/d)	SMM KE
Thomson et al., 2016 [44] Australia	12 weeks	125	Female and male Healthy Age: 61.5 ± 7.4 years. BMI: 27.6 ± 3.6 kg/m^2^	3 days/week 3 sets × 8–12 reps 5 exercises 8 RM	27 g dairy protein spread to each main meal or post RET	P: 1.06 ± 0.10 kg/d (1.42 ± 0.14 kg/d) CON: 1.10 ± 0.10 kg/d	LBM KE
Tieland et al., 2012 [31] Netherlands	24 weeks	62	Female and male Prefrailty and frailty Age: 78 ± 1 years.	2 days/week 3–4 sets × 15–20 reps 6 exercises 50–75% 1RM	30 g milk protein (15 g at breakfast and lunch)	P: 1.0 kg/d (1.3 kg/d) CON: 1.0 kg/d	LBM KE
Verdijk et al., 2009 [29] Netherlands	12 weeks	26	Male Healthy Age: 72 ± 2 years. BMI: 26~27 kg/m^2^	3 days/week 4 sets × 8 reps Leg extension and leg press 75–80% 1RM	20 g casein before and post RET	Both groups (unchanged) 1.1 ± 0.1 kg/d (unchanged)	LBM KE LP
Yamada et al., 2019 [47] Japan	12 weeks	112	Female and male sarcopenic or dynapenic Age: 84.2 ± 5.5 years. BMI: 22 kg/m^2^	2 days/week 7 exercises 3 sets × 20 reps	10 g whey protein, daily	N/A	AMM KE

AMM = appendicular muscle mass; BMI = body mass index; reps = repetitions; CHO = carbohydrate; CON = control group; FFM = fat free mass; KE = knee extension strength; LLMM = lower limb muscle mass; LMB = lean body mass; LP = leg press strength; N/A = not available; P = milk-based protein group; Protein intake: baseline (end of intervention); RET = resistance exercise training; RM = repetition maximum; SMM = skeletal muscle mass.

## Data Availability

No new data were created or analyzed in this study. Data sharing is not applicable to this article.

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
