# Peer review of "Effects of Milk Protein in Resistance Training-Induced Lean Mass Gains for Older Adults Aged ≥ 60 y: A Systematic Review and Meta-Analysis"

_nutrients, 2021, doi:10.3390/nu13082815_

Round 1
Reviewer 1 Report
Nutrients peer review July 2021
The authors present a systematic review and meta-analysis of milk protein and resistance training for improving muscle mass and strength in older adults. With a lot of contradictroy research in this field, the authors attempt to hone in on milk protein with well defined eligibility criteria. The authors look at dosing of milk protein and compare milk protein to whey protein. Interestingly, they note the importance of only supplementing those who are below optimal levels of protein intake at baseline.
General comments:
I am concerned about the search strategy not including ‘older’, which is one of the most commonly used phrases in ageing research. I have noted for example, that Nakayama et al. 2019 was not included in the paper (https://pubmed.ncbi.nlm.nih.gov/32524231/ ) , and neither was Ottesad et al. https://pubmed.ncbi.nlm.nih.gov/29188875/ though perhaps these were excluded because of another exclusion criteria?
I would suggest removing ‘senior’ from the title and throughout the paper. It is not necessary when you already state the age in the title, and typically the phrase used is ‘older adults’.
Specific Comments:
The title focuses on whey versus milk protein, when the paper is really more about milk protein, and the comparison is only a sub-analysis. Consider changing.
Abstract: I would include the key finding of not supplementing those who already have optimal intake, as this is one of the most interesting results of the analysis.
Ln 28/29 - do these studies cited show a direct causation of reduced muscle mass/strength on quality of life? Or an association… Consider rephrasing this to make clear
Ln 30-31 insinuates that those over 90 do not see a benefit. Is there evidence for this?
Reference 10 is 31 years old, do you have newer references for this?
Ln 37 – what does training status mean? Please make this clear.
Paragraph 1 of the introduction is hard to follow, consider rephrasing and making the point about other meta-analyses more explicit.
Ln 45 - the use of therefore here is confusing. This implies that the reason milk is considered superior is due to digestion/absorption property of the protein…?? Please clarify.
Ln 48 suggests whey protein is a compoment of milk protein, this needs clarifiying, as you are comparing milk protein to one of its components? Line 390 is the first time this is made clear, but it should be described clearly in the introduction.
Ln 52 - is ‘training habit’ referring to exercise? Resistance exercise specifically?
Ln 59-60 - what is the background or rationale for this comparison of milk protein to whey protein?
Ln 157 refers to studies removed due to insufficient data but this is not seen in Figure 1
Fig 1 refers to a number of other reasons why studies were removed but these are not discussed in section 3.1. Please clarify.
Table 2: title needs editing
Table 2: could the sample size be in a separate column for ease of viewing?
Ln 290 - consider rephrasing ‘late life’ which insuates extremes of age, when actually the reference cited suggests it decreases with age, rather than specifically referring to ‘late life’ which is somewhat vague.
Ln 294-295 - please clarify what exactly milk protein is superior to.
Ln 313/Ln335 - Suggest referring to the important ESPEN paper when discussing optimal protein intake cut offs: https://www.espen.org/files/PIIS0261561414001113.pdf
Ln358 – this suggestion seems like a jump, do the authors mean to suggest that the protein may not have been digested? This point needs clarifying.
Ln 387/388 - This point about older adults who are dying seems a bit silly, as it would not be clinically indicated to improve muscle mass or strength for someone who is dying. Do the authors actually mean frail or multi-morbid older adults?
Ln 393 – the description here implies that whey protein is effective, albeit less effective than milk protein, but the data suggests the effect of whey protein is not significant in this analysis. Please review.
Ln 407-421 is all repetition, consider removing.
Author Response
The reviewers have provided useful suggestions, which serves to improve the manuscript. We believe the revised manuscript is at good quality.

Reviewer 2 Report
It's a strictly set and well-written review that showed that milk-based protein supplementation may augment resistance training-induced lean body mass or fat-free mass gains and muscle strength for senior adults in a more effective than whey protein, by intake of two cups of milk a day which provides 16 g of milk protein, combined with resistance training.
Author Response
We thank the reviewer's suggestion, which is very helpful. We have revised the manuscript accordingly. The revised manuscript should be at good quality.

Round 2
Reviewer 1 Report
The authors have made the suggested improvements.